# Lower Nerve Growth Factor Levels in Major Depression and Suicidal Behaviors: Effects of Adverse Childhood Experiences and Recurrence of Illness

**DOI:** 10.3390/brainsci13071090

**Published:** 2023-07-18

**Authors:** Michael Maes, Muanpetch Rachayon, Ketsupar Jirakran, Pimpayao Sodsai, Atapol Sughondhabirom

**Affiliations:** 1Sichuan Provincial Center for Mental Health, Sichuan Provincial People’s Hospital, School of Medicine, University of Electronic Science and Technology of China, Chengdu 610072, China; 2Key Laboratory of Psychosomatic Medicine, Chinese Academy of Medical Sciences, Chengdu 610072, China; 3Department of Psychiatry, Faculty of Medicine, King Chulalongkorn Memorial Hospital, Chulalongkorn University, The Thai Red Cross Society, Bangkok 10330, Thailand; muanpetch.mp@gmail.com (M.R.); ket.kett@hotmail.com (K.J.); atapol.s@gmail.com (A.S.); 4Department of Psychiatry, Medical University of Plovdiv, 4002 Plovdiv, Bulgaria; 5Research Institute, Medical University of Plovdiv, 4002 Plovdiv, Bulgaria; 6Kyung Hee University, 26 Kyungheedae-ro, Dongdaemun-gu, Seoul 02447, Republic of Korea; 7Maximizing Thai Children’s Developmental Potential Research Unit, Department of Pediatrics, Faculty of Medicine, Chulalongkorn University, Bangkok 10330, Thailand; 8Center of Excellence in Immunology and Immune-Mediated Diseases, Department of Microbiology, Faculty of Medicine, Chulalongkorn University, Bangkok 10330, Thailand; yokpim@gmail.com

**Keywords:** major depression, mood disorders, inflammation, neuro-immune, biomarkers

## Abstract

Major depressive disorder (MDD) and its severe subtype, major dysmood disorder (MDMD), are distinguished by activation of inflammatory and growth factor subnetworks, which are associated with recurrence of illness (ROI) and adverse childhood experiences (ACEs). Nerve growth factor (NGF) plays a crucial role in facilitating neuro-immune communications and may regulate the inflammatory response. Methods: The present study examined the effects of ACEs and ROI on culture supernatant NGF, stem cell factor (SCF), stem cell GF (SCGF), hepatocyte GF (HGF), and macrophage colony-stimulating factor (M-CSF), in relation to a neurotoxicity (NT) cytokine profile. Results: NGF levels are lower in MDD (*p* = 0.003), particularly MDMD (*p* < 0.001), as compared with normal controls. ROI and ACE were significantly and inversely associated with NGF (≤0.003) and the NGF/NT ratio (≤0.001), whereas there are no effects of ACEs and ROI on SCF, SCGF, HGF, or M-CSF. Lowered NGF (*p* = 0.003) and the NGF/NT ratio (*p* < 0.001) are highly significantly and inversely associated with the severity of the current depression phenome, conceptualized as a latent vector extracted from the current severity of depression, anxiety, and suicidal behaviors. We found that one validated and replicable latent vector could be extracted from NGF, ROI, and the depression phenome, which therefore constitutes a novel ROI-NGF-pathway-phenotype. ACEs explained 59.5% of the variance in the latter pathway phenotype (*p* < 0.001). Conclusions: The imbalance between decreased NGF and increased neurotoxic cytokines during the acute phase of severe depression may contribute to decreased neuroprotection, increased neuro-affective toxicity, and chronic mild inflammation.

## 1. Introduction

The acute phase of major depressive disorder and a major depressive episode (MDD/MDE) are distinguished by the activation of the immune-inflammatory response system (IRS) and the compensatory immune-regulatory system (CIRS) [1]. The activation of the IRS is indicated by elevated levels of M1 cytokines, such as interleukin (IL)-1β, IL-6, and tumor necrosis factor (TNF)-α; T helper (Th)1 cytokines, such as IL-2 and IFN-γ; and Th17 cytokines, such as IL-17 [1]. Elevated concentrations of Th2 cytokines, specifically IL-4, and T regulatory (Treg) cytokines, particularly IL-10, are indicative of CIRS activation in MDD/MDE. The activation of the IRS is typically associated with an acute phase or inflammatory response, as well as T cell activation [2,3]. The latter is characterized by an upregulation of T cell activation markers, such as CD71 and CD40L [4]. CIRS activation, which preserves immune homeostasis or tolerance, is indicated by elevated levels of certain acute phase proteins (e.g., haptoglobin), soluble IL-1 receptor antagonist (sIL-1RA), and sIL-2R [1]. It is worth noting that the acute phase of MDD/MDE is marked by increments in neurotoxic M1, Th1, Th2, and Th17 cytokines because of IRS activation [5]. The constituents of the IRS and neurotoxic immune profiles have been enumerated in Appendix A. The observed impact on neural and astroglial projections, along with other neuronal functions, has led to the hypothesis that the neurotoxic profile in question may contribute to heightened neuro-affective toxicity, potentially resulting in depression [1,6].

Maes et al. [5] have reported an increase in the stimulated production of growth factors, namely platelet-derived growth factor (PDGF), vascular endothelial growth factor (VEGF), and basic fibroblast growth factor (FGF2), in individuals with MDD as compared to those without the disorder. The growth factors associated with MDD, along with their respective gene IDs and nomenclature, are provided in Appendix A. The neurotoxic cytokines, as well as these three growth factors, interact during depression in a well-defined protein–protein interaction (PPI) network, which comprises two subnetworks, namely an IRS/CIRS immune and a growth factor subnetwork [5]. Moreover, the activated growth factor profile was significantly associated with the neurotoxicity profile and both current suicidal behaviors and the depression phenome [5]. Thus, it was deduced that the growth factors are responsible for inducing or sustaining neurotoxicity associated with IRS/CIRS, thereby contributing to the pathophysiology of depression [5].

However, it is unknown if other growth factors, such as nerve growth factor (NGF), stem cell factor (SCF), stem cell growth factor (SCGF), hepatocyte growth factor (HGF), and macrophage colony stimulating factor (M-CSF), are altered in depression or predict the depression phenome or suicidal behaviors. Appendix A illustrates the PPI network of these five growth factors, and Table 1 shows the most important gene ontologies (GO), biological processes and molecular functions (Gene Ontology overview), WIKIpathways (Home|WikiPathways), and the Kyoto Encyclopedia of Genes and Genomes (KEGG) pathways (KEGG PATHWAY Database (genome.jp) that were over-represented in this PPI network. Additionally, as shown in Appendix A, PDGF, VEGF, and FGF2 are all part of the same tight growth factor network, indicating that there are numerous interactions between these eight growth factors. Nevertheless, a previous meta-analysis showed that serum NGF is lower in MDD as compared with controls [7]. Moreover, animal models of depression show lowered central NGF levels and indicate that stressors decrease NGF levels [8].

We have recently found that number of prior depressive episodes and suicidal attempts, as well as the severity of lifetime suicidal ideation, are associated with increasing sensitization of the IRS, neurotoxicity, and growth factor profiles [9,10]. The reoccurrence of illness (ROI), conceptualized as a latent vector extracted from the reoccurrence of depressive episodes and suicidal behaviors, predicts not only those immune profiles but also the depression phenome including current suicidal behaviors, either ideation or attempts [9,10]. In addition, a validated latent vector can be extracted from the ROI, IRS, neurotoxicity, and growth factor profiles [11]. As a result, we were able to develop a ROI-immune-growth-factor pathway phenotype that accurately predicts the severity of the depression phenome [11].

Adverse childhood experiences (ACEs), which include mental neglect and trauma, physical trauma, domestic violence, and the loss of a parent, are strongly associated with increasing ROI; lifetime and current suicidal behaviors; the depression phenome; and elevated IRS, neurotoxic, and GF profiles [11]. In addition, the effects of ACEs on the depression phenome, which includes suicidal behaviors, are entirely mediated by the ROI-pathway-phenotype [11]. These findings suggest that the accumulation of ACEs sensitizes the immune and growth factor subnetworks and the ROI and plays a role in the development of depression, disease severity, and increased suicidal behaviors [11]. Despite this, it is unknown if ACEs and ROI are associated with the stimulated production of NGF, SCF, SCGF, HGF, and M-CSF and if alterations in these growth factors mediate the effects of ACEs on the depression phenome. Accordingly, we are particularly interested in NGF in relation to other growth factors and the neurotoxicity profile for the following reasons: (a) NGF belongs, as brain-derived neurotrophic factor (BDNF), to the neurotrophin family, and may play a role in psychiatric and neurodegenerative disorders by modulating neuronal activation and survival, brain plasticity, behaviors, and neurocognition [12,13,14]; and (b) NGF is a key element in neuro-immune communication between the neurological and immune systems, and it may modulate the inflammatory response, acting as both a pro- and anti-inflammatory protein [15,16].

Hence, the present study was carried out to examine: (a) NGF, SCF, SCGF, HGF and M-CSF, in relation to VEGF, PDGF and FGF, and the neurotoxicity immune profile in MDD; and (b) whether NGF or the other growth factors mediate the effects of ACEs and ROI on the depression phenome. The specific hypotheses are that there is a relative decline in NGF with respect to the neurotoxicity, immune, and growth factor profiles, and that ACEs and ROI are associated with a relative decline in NGF.

## 2. Methods and Participants

### 2.1. Participants

Thirty patients with major depression were recruited from the outpatient clinic of the Department of Psychiatry at King Chulalongkorn Memorial Hospital in Bangkok, Thailand for this study. We recruited 20 healthy participants of both sexes between the ages of 18 and 65 from the same catchment area, Bangkok, Thailand. The control group was recruited through word-of-mouth, and controls with a diagnosis of any DSM-5 axis 1 condition or a positive family history of MDD or BD were excluded from the study. The patients were diagnosed with MDD based on DSM-5 criteria [17] and had moderate to severe depression as determined by a score of 17 or higher on the Hamilton Depression Rating Scale (HAMD) [18]. Moreover, using the precision nomothetic psychiatry approach [9,10], we divided the MDD group into those with major dysmood disorder (MDMD) versus simple DMD (SDMD). MDMD is characterized by increased ACEs, ROI, suicidal behaviors, and phenome scores, while MDMD, and not SDMD, shows aberrations in multiple biomarkers and pathways, including neurotoxicity and growth factor profiles [9,10].

Participants with MDD were excluded if they suffered from other DSM-5 axis 1 disorders, including obsessive-compulsive disorder, substance use disorders, psycho-organic disorders, schizophrenia, schizoaffective disorders, and post-traumatic stress disorder. MDD patients and healthy controls were excluded if they had (auto)immune diseases, including cancer, chronic obstructive pulmonary disease, type 1 or type 2 diabetes mellitus, psoriasis, inflammatory bowel disease, or asthma; neurological, neuroinflammatory, and neurodegenerative disorders such as epilepsy, stroke, multiple sclerosis, or Alzheimer’s or Parkinson’s disease; and inflammatory or allergic reactions three months prior to the study. In addition, we excluded participants who had received immunomodulatory drugs (lifetime history) such as glucocorticoids, therapeutic doses of omega-3 or antioxidant supplements, or anti-inflammatory medication in the month preceding the study. We also excluded pregnant and lactating women. We statistically accounted for the potential effects of the patients’ drug use, including benzodiazepines (n = 22), atypical antipsychotics (n = 14), sertraline (n = 18), other antidepressants (n = 8, including fluoxetine, venlafaxine, escitalopram, bupropion, and mirtazapine), and mood stabilizers (n = 4). Nevertheless, no significant drug effects on GF could be detected.

Before participating in this study, all subjects and controls provided written consent. The investigation was conducted in accordance with international and Thai ethical standards and privacy laws. The institutional Review Board of Chulalongkorn University’s Faculty of Medicine in Bangkok, Thailand, approved the study (#528/63) in accordance with the International Guidelines for the Protection of Human Subjects as required by the Declaration of Helsinki, the Belmont Report, the CIOMS Guideline, and the International Conference for Harmonization of Good Clinical Practice (ICH-GCP).

### 2.2. Clinical Measurements

A research assistant specialized in affective disorder research conducted interviews that were semi-structured and included the number of depressive episodes. The Mini-International Neuropsychiatric Interview was used to evaluate axis-1 diagnoses [19]. We used the HAMD, 17-item version [18], administered by a senior psychiatrist to evaluate the severity of depression. The Thai State-Trait Anxiety Assessment (STAI) is an inventory designed to measure the intensity of state anxiety [20]. The lifeline version of the Columbia-Suicide Severity Rating Scale (C-SSRS) [21] was used to evaluate lifetime (LT) and current suicidal behaviors (SBs). The LT and current SBs were calculated as previously described [22,23]. LT_SB was conceptualized as a principal component (PC) derived from the C-SSRS items that reflect LT suicidal ideation and attempts [22,23]. Current_SB was conceived as a PC derived from items reflecting current ideation and attempts [22,23]. All PCs that we constructed complied with specific quality criteria in terms of convergence, construct validity, and replicability (see also Statistics). To compute the ROI, we utilized the number of depressive episodes, C-SSRS item 1 (lifetime suicidal ideation, namely the desire to die), and the C-SSRS item lifetime number of actual attempts and extracted one PC that met the quality criteria. The depression phenome (PC_phenome) was determined as the first PC extracted from current SB, HAMD, and STAI scores [22,23]. Using DSM-5 criteria, tobacco use disorder (TUD) was diagnosed. Body mass index (BMI) was determined by dividing body weight (in kilograms) by length squared (in meters).

### 2.3. Assays

After a 10-h overnight fast, blood was drawn at 8:00 a.m. in BD Vacutainer^®^ EDTA (10 mL) vials (BD Biosciences, Franklin Lakes, NJ, USA). In this investigation, we measured the growth factors in stimulated whole blood culture supernatant as previously described [5,11,24]. We utilized RPMI-1640 medium (Gibco Life Technologies, Indianapolis, IN, USA) supplemented with L-glutamine and phenol red and containing 1% penicillin (Gibco Life Technologies, Indianapolis, IN, USA), 5 µg/mL PHA (Merck, Germany), and 25 µg/mL lipopolysaccharide (unstimulated) (LPS; Merck, Germany). On sterile 24-well plates, 1.8 mL of each of these two mediums was added to 0.2 mL of 1/10-diluted whole blood. The specimens from each subject were incubated for 72 h at 37 °C, 5% CO_2_ in a humidified atmosphere. After incubation, the plates were centrifuged at 1500 rpm for 8 min. The supernatants were removed meticulously under sterile conditions, divided into Eppendorf tubes, and immediately frozen at −70 °C until thawed for cytokine/growth factor assays. Using a multiplex method, the cytokines/growth factors were quantified using the LUMINEX 200 apparatus (BioRad, Carlsbad, CA, USA). In brief, supernatants were diluted fourfold with medium and incubated with magnetic beads for 30 min. After 30 min and 10 min, respectively, after adding detection antibodies and streptavidin-PE, the fluorescence intensities (FI) were measured. We utilized the FI values (blank subtracted) for statistical analyses in the current study, as FI are frequently preferable to absolute concentrations [5]. All FI values fell within the concentration curve, allowing quantification of all cytokines. In all investigations, the inter-assay CV values are less than 11%.

Appendix A lists the names, abbreviations, and official gene symbols of all cytokines/growth factors measured in this study. Appendix A provides a listing of the growth factor, neurotoxicity, and other profiles investigated here, and shows the z unit-based composite scores measured in this study, namely zNGF-zNT (reflecting the NGF/neurotoxicity ratio), zNGF-zGF (reflecting the ratio of NGF/PDGF + FGF + VEGF), and zNGF-z(NT + GF).

### 2.4. Statistical Analysis

Scale variables were compared using analysis of variance (ANOVA) or the Kruskal–Wallis test, while nominal variables across categories were compared using chi-square tests or Fisher’s Freeman–Halton test. Using Pearson’s product–moment correlation coefficients, correlations between two scale variables were analyzed. Using multiple regression analysis (manual method and automatic method with a *p*-to-entry of 0.05 and a *p*-to-remove of 0.06 while evaluating the change in R^2^), we determined the biomarkers that predict SB or phenome scores. Collinearity issues were checked with tolerance and VIF, multivariate normality with Cook’s distance and leverage, and homoscedasticity with the White and modified Breusch–Pagan tests. The results of these regression analyses were always bootstrapped utilizing 5000 bootstrap samples, and if the results were not congruent, the bootstrapped results are presented. Principal component analysis (PCA) was utilized to summarize the data in summary indices or patterns. When the first principal component explained at least 50.0% of the total variance and all loadings were greater than 0.7, it was deemed to indicate a valid pattern. In addition, factorability was estimated using the Kaiser–Meyer–Olkin (KMO) test for sampling adequacy (should be >0.6) and Bartlett’s test for sphericity. All statistical analyses (except PLS) were performed using version 28 of IBM SPSS Windows.

Using SmartPLS path analysis [25], we assessed the causal relationships between ACEs, ROI, growth factors, and the depression phenome. The latter was conceptualized as a latent vector derived from Current_SB and the HAMD and STAI scores. Other indicators were submitted as single indicators. Complete PLS analysis was conducted when the outer and inner models satisfied predefined quality criteria, namely: (a) all latent vectors have adequate average variance extracted (AVE) (>0.50); Cronbach’s alpha (>0.7), composite reliability (>0.7), and rho A (>0.8) scores were present; (b) all loadings on the latent vectors are greater than 0.6 at *p* < 0.001; (c) using Confirmatory Tetrad Analysis, the models are specified as reflective models; (d) the model fit SRMR is less than 0.08; (e) discriminant validity as checked using the Heterotrait-Monotrait ratio (HTMT) is established; and (f) model replicability is adequate using PLSpredict and Q2 predict and the cross-validated predictive ability test (CVPAT). Using 5000 bootstrap samples, complete PLS analysis was performed in order to determine total direct and indirect effects, specific indirect effects, and path coefficients (with exact *p*-values). The level of statistical significance was set at *p* < 0.05 for two-tailed tests.

The primary statistical analysis was a multiple regression with the depressive phenotype as the dependent variable. A priori power analysis (G*Power 3.1.9.4) for a linear multiple regression analysis with an effect size of 0.25 (equivalent to around 20% explained variance), alpha = 0.05, power = 0.8, and 3 covariates suggests a minimum sample size of 48.

Using seed-genes, we constructed protein–protein interaction (PPI) networks using differentially expressed proteins. The PPI network was built using version 11.0 of the predictive database STRING (https://string-db.org, accessed on 10 February 2023). With the organism set to Homo sapiens and a minimum interaction score requirement of 0.400, we derived zero-order PPIs. The enrichment scores and annotated terms of the growth factor PPI network were analyzed using the STRING-provided tools, namely: GO biological processes and molecular functions, (Gene Ontology overview), WIKIpathways (Home|WikiPathways) and the Kyoto Encyclopedia of Genes and Genomes (KEGG) pathways (KEGG PATHWAY Database (genome.jp). Enrichment analysis results are always presented with false discovery rate (FDR)-adjusted *p*-values.

## 3. Results

### 3.1. Demographic and Clinical Data of the Study Groups

Table 2 shows the demographic and clinical data of the healthy controls and SDMD and MDMD patients. There were no significant differences in age, sex, education, or TUD among the study groups. Depressed patients showed a somewhat higher BMI than healthy controls. The prevalence of subjects with melancholic and psychotic features was significantly higher in MDMD than in SMDM. The HAMD, STAI, and ACE scores (and prevalence of separate ACEs) were significantly higher in patients than controls. The number of episodes, LT_SB, ROI, Current_SB, and depression PC_phenome scores were higher in MDMD than is SDMD and controls.

### 3.2. Growth Factors in Depression

Table 3 shows the measurements of the growth factors and the ratios in MDD versus controls. We found z_NT and z_GF were significantly higher in subjects with depression than in control subjects. NGF, the zNGF-zNT and zNGF-zGF ratios were significantly lower in patients with depression than control subjects. Those differences remained significant after FDR *p* correction, except for zNT (*p* = 0.06).

Figure 1 and Appendix A show the measurements of the growth factors in MDMD, SDMD, and controls. We found that NGF, zNGF-zNT, and zNGF-z(NT + GF) were significantly lower in MDMD than in controls, whereas there were no differences between controls and SDMD. The zNGF-zGF was highly significantly different (*p* < 0.001) between MDMD and controls, whereas the differences between SDMD and controls were only marginally significant (*p* = 0.042).

### 3.3. Intercorrelation Matrix

Table 4 shows the intercorrelation matrix between the growth factor variables and ACEs, ROI, Current_SB and PC_phenome data. ACEs, ROI, Current_SB and PC_phenome were significantly and inversely associated with NGF, zNGF-zNT, zNGF-zGF, and zNGF-z(NT + GF). Figure 2 shows the partial regression of zNGF-z(NT + GF) on ACEs and Figure 3 shows the partial regression of NGF on ROI.

### 3.4. Results of Multiple Regression Analysis with Clinical Data as Dependent Variables

Table 5 shows the results of multiple regression analysis with clinical variables as dependent variables and growth factor data as explanatory variables while allowing for the effects of age, sex, education, TUD, and BMI. Table 5 regression #1 shows that 15.9% of the variance in Current_SB was explained by the zNGF-zGF ratio. Regression #2a and #2b show that 19.7% and 18.9% of the variance in LT_SB and LT + Current_SB, respectively, was explained by zNGF-zNT. Regression #3a shows that 38.4% of the variance in PC_phenome was explained by zNGF-z(NT + GF) and age (both inversely). We have rerun the same analysis and entered the ACE data (regression #3b). The latter multivariate regression shows that 70.8% of the variance in PC_phenome was explained by zNGF-zNT and age (both inversely) and mental and sexual trauma (positively). Figure 4 shows the partial regression of PC_phenome on the zNGF-z(NT + GF) score.

### 3.5. Results of PLS Analysis

Before conducting PLS analysis, we examined whether one PC could be extracted from NGF data and the ROI and depression PC_phenome scores. We detected that one validated PC could be extracted from PC_phenome (loading = 0.854), ROI (0.867), zNGF-zNT (−0.851), and zNGF-zGF (−0.843) (KMO = 0.615, Bartlett’s test of sphericity: χ^2^ = 200.22, df = 4, *p* < 0.001, VE = 72.88%). As such, we were able to construct a ROI-NGF-pathway-phenotype. Figure 5 shows the results of PLS-SEM analysis with ACEs as input variable, the above latent vector as mediator, and the diagnosis (controls, SDMD and MDMD, entered as ordinal variables) as output. Complete PLS bootstrapping (5000 samples) showed that the model was adequate with SRMR = 0.049, all outer loadings > 0.787, AVE = 0.719, Cronbach’s alpha = 0.873, composite reliability = 0.911, and rho_A = 0.910). The model showed significant replicability (PLSpredict), with all Q^2^ values being between 0.243–0.577 and the CVPAT indicator average for all indicators being significant (*p* < 0.0001). We found that 59.5% of the variance in the ROI-NGF-pathway-phenome was explained by ACEs. Nevertheless, this model is not perfect, as no discriminatory validity could be established between the ROI-NGF-pathway-phenotype and the diagnosis (HTMT = 0.930). In fact, the ordinal diagnosis variable could be included into the ROI-NGF-pathway-phenotype latent vector (all loadings > 0.754, AVE = 0.744, Cronbach’s alpha = 0.913, composite reliability = 0.935, rho_A = 0.935). ACEs explained 61.1% of the variance in this pathway phenotype.

## 4. Discussion

### 4.1. NGF in MDD/MDMD, Suicidal Behaviors and ROI

The first main conclusion of this study is that NGF levels are much lower in MDD, particularly in MDMD, and that the ratios of NFG to the immune-associated neurotoxicity index are significantly lower in both depression subtypes. However, whereas NGF was greatly reduced in depression/MDMD, other growth factors (SCF, SCGF, HGF, M-CSF) were unchanged, while VEGF, PDGF, and FGF were increased. These findings build on a previous systematic review and meta-analysis of seven studies that found reduced NGF levels in MDD patients relative to controls [7]. Nevertheless, the current investigation found that such NGF declines are more common in MDMD than SDMD. Because MDMD is the more severe subtype, our findings are consistent with those of Chen et al. [7], who discovered decreased NGF levels in the most severe MDD subgroup when compared to controls (ES: 0.50, 95% CI: 0.83 to 0.18, *p* = 0.002). Lower NGF levels have also been observed in the hippocampus in some animal models of depression, but not in the Flinders Sensitive Line rat model [26,27,28]. Reduced hippocampus NGF as a result of folate shortage may lead to depressive and anxious behaviors [29].

In our investigation, we discovered that lower NGF levels were inversely related to the intensity of the depressive phenome, which included depression, anxiety, and suicidal behaviors (rated during the previous month). Our findings support the findings of a previous meta-analysis, suggesting that low NGF levels are negatively related to illness severity [7], while a study in suicide victims showed that NGF mRNA was downregulated in the hippocampus [30]. Nevertheless, our investigation discovered that lower composite scores of zNGF-zNT or zNGF-zGF indices were far stronger predictors of the intensity of the depressive phenome than NGF.

Surprisingly, whereas depression is related to lower peripheral NGF levels, more “positive” emotional states, such as love, are associated with higher NGF levels [31]. Moreira et al. [32] discovered that serum NGF concentrations were higher in affective patients transitioning to bipolar disorder as compared to current or remitted MDD phenotypes. Interestingly, this transition was not accompanied by changes in other neurotrophic factors (BDNF or glial cell-derived neurotrophic factor) or immune–inflammatory mediators (IL-6 and TNF-α). As a result, NGF levels appear to be a sensitive indicator of the emotional state of depression and suicidal behaviors as well.

### 4.2. NFG Is Strongly Associated with ROI

The second major finding is that increasing ROI was associated with a reduction in NGF, zNGF-zNT, and zNGF-zGF indices. The ROI is a crucial characteristic of affective disorders because it reflects the recurrence of episodes and suicidal behaviors and predicts the severity of depression, including current suicidal behaviors [9,10,23]. Moreover, we observed that ROI is significantly associated with increases in the cytokine and GF (VEGF, PLGD, and FGF) networks, and increased immune neurotoxicity [5]. Furthermore, we found that ROI is associated with lowered antioxidant defenses, including the high-density-lipoprotein-PON1 complex, increased oxidative stress [23], and the gut microbiome [33]. In addition, one factor could be extracted from ROI and the GF (VEGF, PDGF, and FGF) and the neurotoxicity profile, forming a novel pathway–phenotype that predicts the phenome of depression [11] Such findings demonstrate that ROI and GF/neurotoxicity pathways are intricately intertwined and comprise a crucial aspect of the disease.

In fact, the present study extended these findings by demonstrating that one valid latent vector could be extracted from ROI, zNGF-zNT, and zNGF-zGF indices, and the depression phenome. This suggests that ROI, NGF indices, and the phenome are manifestations of the same underlying core, namely the severity of the illness depression. Consequently, NGF and its indices are integral components of the severity of recurrent depression.

### 4.3. Effects of ACEs on NGF

The third major finding of this study is that there are significant inverse associations between the total sum of ACEs and NGF levels as well as the zNGF-zNT and zNGF-zGF indices, whereas there were no effects of ACEs on SCF, SCGF, HGF, or M-CSF. Previously, we observed that ACEs are associated with elevated GF and neurotoxicity profiles, explaining that ACEs account for a substantial proportion (28–33%) of the composite’s variance. Furthermore, ACEs predicted approximately 59.6% of the variance in the ROI-NGF-pathway-phenome score, indicating that childhood psychosocial stressors reduce NGF and are strongly linked to the core of the illness.

While mild transient stressors (examination stress) have no effect on NFG serum concentrations in humans [34,35], there are many studies indicating that stress in animal models decreases NGF. In the rodent model of stress-induced helplessness, NGF levels were temporarily reduced by a quarter six hours after the stress treatment [36]. Interestingly, in the latter experiment, BDNF levels were not altered, and BDNF levels in the right frontal cortex were three times higher than in the left. Chronic mild stress, but not acute stress or prenatal stress, decreased the mature form of NGF and increased proNGF levels in the hypothalamus of male rats, indicating that chronic stress inhibits NGF maturation or stimulates the degradation of NGF [26]. In the rodent, the frontal cortex and amygdala NGF protein levels decrease substantially following exposure to a novel environment and handling stress [37]. In addition, prolonged immobilization stress decreased NGF mRNA in the rat hippocampus [38] and forced motoric activities or rotatory stress reduces NGF levels in the hippocampus [37,39]. On the other hand, cold stress (one or more episodes) may increase mRNA levels of NGF in the hippocampus [40], and highly arousing conditions may increase plasma NGF [41]. A recent review concludes that NGF plays a crucial role in stress responses by translating environmental noxious stimuli into multiple feedback systems under both physiological and pathological conditions [42].

### 4.4. Lowered NGF and the Pathophysiology of Depression

Our findings indicate that (a) decreased peripheral NGF production is involved in the pathophysiology of depression and at least partially mediates the effects of ACEs on the phenome of depression; and (b) the decreases in NGF relative to increases in neurotoxic immune products and other growth factors that stimulate the cytokine network indicate an imbalance between neuroprotective and detrimental pathways. The pathophysiological role of NGF in the neuro-immune system can be explained by its function as a neurotrophin and, therefore, a neuroprotective factor, its immune-regulatory effects, and its influence on the neuro-immune cross-talks.

First, NGF is a neurotrophin that regulates neuronal growth, apoptosis, neuronal survival and repair after injury, and neuronal plasticity [16,43,44,45]. The neurotrophic effect is mediated via Trk receptor activation and the P13K/Akt pathway [46] in the hippocampus, where NGF protects against excitotoxic signals leading to neuronal degeneration and cell death. As mentioned previously, chronic moderate stress decreases mature NGF levels in conjunction with decreased TrkA-related survival signaling and enhanced pro-apoptotic signaling via p75NTR [26]. There is now evidence that reduced neurotrophin signaling contributes to the pathophysiology of neurodegenerative diseases [47], whilst decreased hippocampal NGF levels contribute to neurodegeneration associated with depressive and anxiety-like behaviors in folate-deprived animals [29]. Importantly, astroglia may acquire an asthenic and neurotoxic profile in response to reduced ambient NGF levels [48].

Second, NGF has pro-inflammatory effects by activating macrophages, dendritic cells, neutrophils, B cells, T cells, and mast cells, as well as by enhancing the proliferative response of T and B cells [16]. Nevertheless, NGF has profoundly negative immuno-regulatory effects, including the activation of the cholinergic pathway, which inhibits pro-inflammatory cytokine release [16]. As a result, NGF stimulates innate immune responses while simultaneously increasing IL-10 production, dampening the inflammatory response, and limiting tissue injury [16]. NGF may attenuate inflammatory responses via TrkA receptors, and decreased TrkA expression may contribute to chronic inflammation [16]. As a result, the imbalance between decreased NGF and increased neurotoxic cytokines, driven in part by increased GF production during the IRS response, may not only contribute to the maintenance of chronic inflammation, but also to decreased neuroprotection and increased neuro-affective toxicity, and thus affective symptoms and suicidal behaviors.

### 4.5. Limitations

This study would have been more intriguing if we had measured acute negative life events to determine whether the latter could influence NGF signaling independently of ACEs. It would be intriguing to test the TrkA receptor on immune cells from depressed individuals. Future research should investigate whether in vitro administration of NGF can inhibit the neurotoxic cytokine production observed in MDD and MDMD. To determine whether NGF is associated with the onset of depression, it would be necessary to measure NGF during the various phases of depression (acute, partial remission, and remission) and in first-episode depression. One could claim that the sample size is relatively small, resulting in less accurate parameter estimates. Despite this, the research was conducted with a power of 0.8, and post-hoc analysis of the study’s primary analysis (regression of phenome scores on NGF indices) revealed that the actual power was 0.99 (multiple regression analysis, 2 predictors, R^2^ = 0.35).

## 5. Conclusions

The levels of NGF have been observed to be significantly lower in individuals diagnosed with MDD with a more pronounced decrease in those with MDMD when compared to individuals without any psychiatric diagnosis. The study found a significant correlation between an increase in ROI and ACEs with a notable decrease in NGF and zNGF-zNT ratio. Our study reveals that a single latent vector can be extracted from data pertaining to NGF, ROI, and the depression phenotype, and that 59.5% of the variance in this ROI-NGF-pathway phenotype may be explained by ACEs. The findings suggest that a reduction in NGF during the acute phase of severe depression may result in heightened neuro-affective toxicity.

## Figures and Tables

**Figure 1 brainsci-13-01090-f001:**
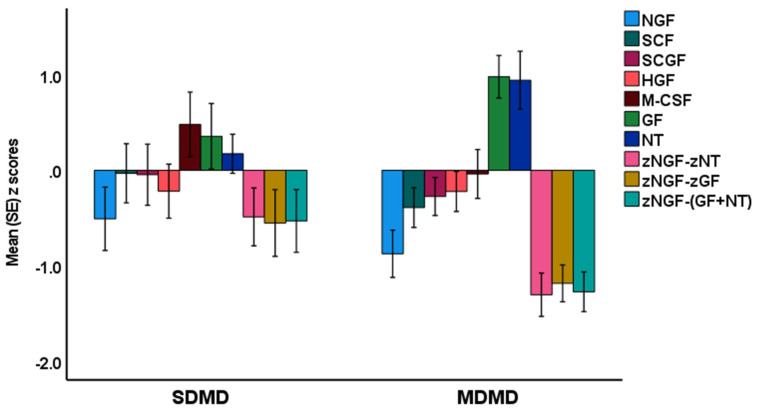
Measurements of the growth factors in major dysmood disorder (MDMD) and simple dysmood disorder (SDMD) as compared with healthy controls (set at 0 value). NGF: nerve growth factor; SCF: stem cell factor; SCGF: stem cell growth factor; HGF: hepatocyte growth factor; M-CSF: macrophage colony stimulating factor. GF: composite built using platelet-derived growth factor, vascular endothelial growth factor, and fibroblast growth factor; NT: neurotoxicity: composite of neurotoxic cytokines.

**Figure 2 brainsci-13-01090-f002:**
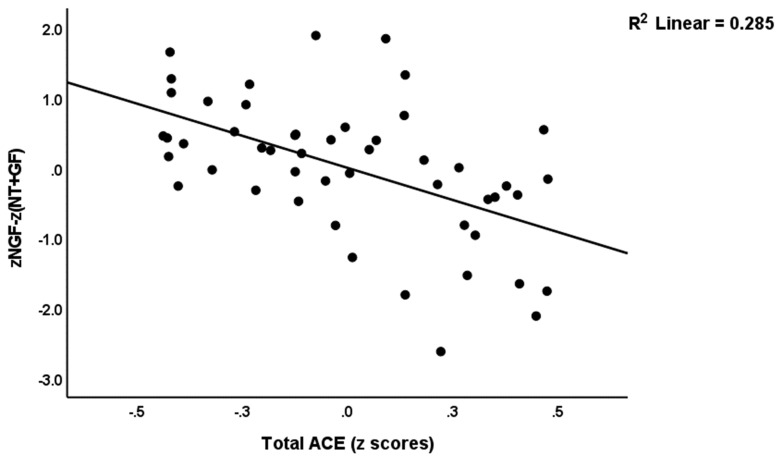
Partial regression of the ratio of nerve growth factor (NGF) on neurotoxicity (NT) and other growth factors [zNGF-z(NT + GF)] on adverse childhood experiences (ACEs).

**Figure 3 brainsci-13-01090-f003:**
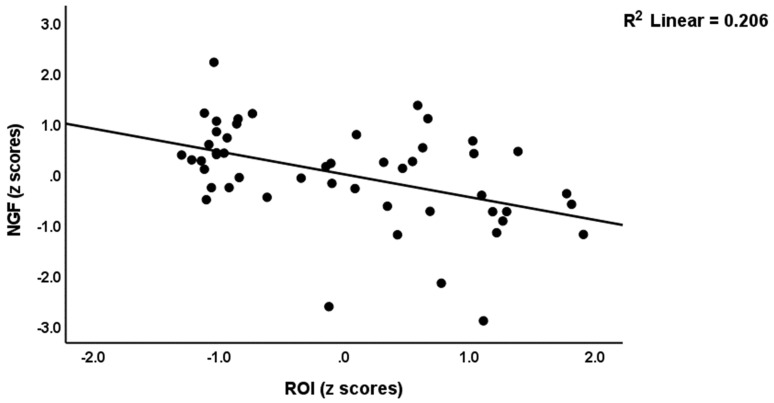
Partial regression of nerve growth factor (NGF) levels on recurrence of illness (ROI).

**Figure 4 brainsci-13-01090-f004:**
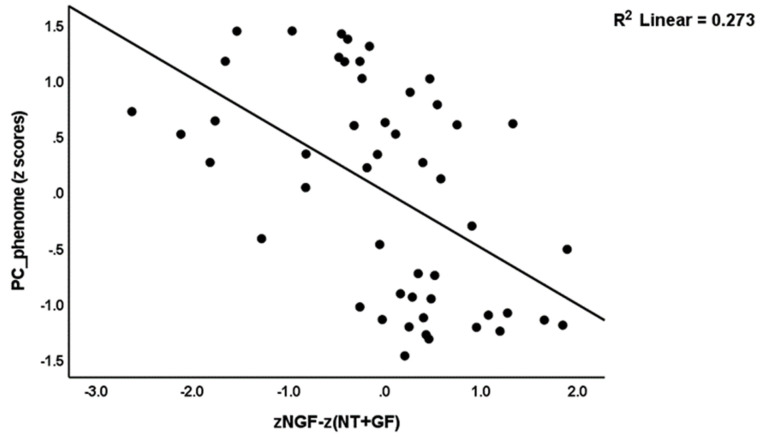
Partial regression of the depression phenome (PC_phenome) score on the ratio of nerve growth factor (NGF) to neurotoxicity (NT) and other growth factors (GF) [zNGF-z(NT + GF)].

**Figure 5 brainsci-13-01090-f005:**
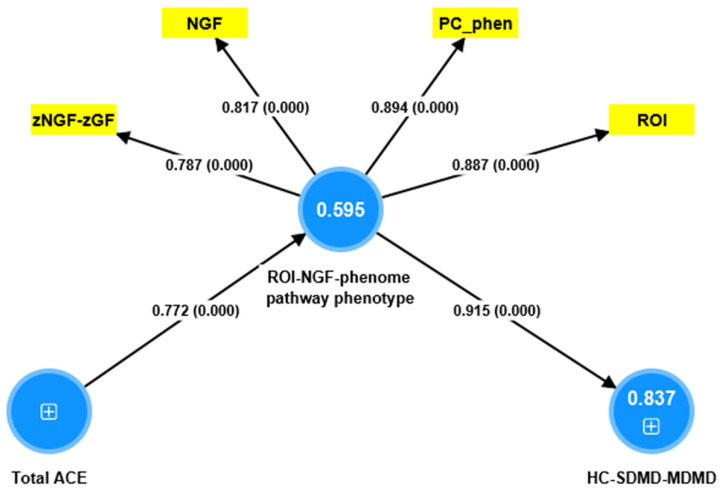
PLS-SEM model. ACE: Adverse childhood experiences, ROI: recurrence of illness, NGF: nerve growth factor, GF: composite built using platelet-derived growth factor, vascular endothelial growth factor, and fibroblast growth factor; PC_phenome: first factor extracted from current phenome data, HS-SDMD-MDMD: healthy controls—simple dysmood disorder—major dysmood disorder (ordinal scale). ACE is entered as the input variable, the ROI-NGF-phenome latent vector as mediator, and the diagnosis (controls, SDMD and MDMD) as the output. Shown are the path coefficients (with *p*-value) and loadings (with *p* values) of the outer model; figures in the blue circles denote explained variance.

**Table 1 brainsci-13-01090-t001:** Molecular functions, KEGG and Wiki pathways enriched in the network of nerve growth factor (NGF), stem cell factor (SCF), stem cell growth factor (SCGF), hepatocyte growth factor (HGF), and macrophage colony stimulating factor (M-CSF).

#Term ID	Term Description	Observed Gene Count	Background Gene Count	Strength	False Discovery Rate
GO:0008083	Growth factor activity	5	161	2.08	1.34 × 10^7^
hsa04014	Ras signaling pathway	4	226	1.84	1.99 × 10^5^
hsa04015	Rap1 signaling pathway	4	202	1.89	1.99 × 10^5^
hsa04010	MAPK signaling pathway	4	288	1.74	2.69 × 10^5^
hsa04151	PI3K-Akt signaling pathway	4	350	1.65	4.36 × 10^5^
WP2848	Pluripotent stem cell differentiation pathway	3	48	2.39	0.00011
WP3932	Focal adhesion: PI3K-Akt-mTOR-signaling pathway	4	302	1.71	0.00011
WP4172	PI3K-Akt signaling pathway	4	336	1.67	0.00011
GO:0005126	Cytokine receptor binding	3	264	1.65	0.0133

GO: gene ontology, molecular functions, hsa: KEGG pathways, WP: Wiki Pathways.

**Table 2 brainsci-13-01090-t002:** Demographic and clinical data of the healthy controls (HC) and depressed patients divided into those with simple depression (SDMD) and major dysmood disorder (MDMD).

Variables	HC ^a^ (n = 20)	SDMD ^b^ (n = 11)	MDMD ^c^ (n = 19)	F/χ^2^/FFHET /KW	df	*p*
Age (years)	33.6 (8.0)	27.0 (5.4)	29.6 (9.9)	2.47	2/47	0.095
Sex (male/female)	6/14	4/7	7/12	0.24	2	0.888
Education (years)	16.1 (2.2)	16.5 (0.9)	15.1 (1.3)	2.99	2/47	0.060
TUD (no/yes)	18/2	9/2	14/5	1.78	-	0.408
BMI (kg/m^2^)	21.33 (2.51)	25.49 (5.55)	25.55 (6.32)	4.32	2/47	0.019
Melancholia–psychosis (no/yes)	20/0 ^c^	11/0 ^c^	13/6 ^a,b^	FFHET	-	0.003
HAMD	0.9 (1.5) ^b,c^	22.2 (5.7) ^a^	24.3 (5.8) ^a^	147.01	2/47	<0.001
STAI	37.7 (10.6) ^b,c^	56.8 (5.2) ^a^	56.9 (8.2) ^a^	28.00	2/47	<0.001
Total ACEs	13.9 (4.6) ^b,c^	25.6 (4.5) ^a^	29.5 (7.3) ^a^	38.40	2/47	<0.001
Mental trauma	19/1	5/6	8/11	FFHET	-	<0.001
Physical trauma	19/1	6/5	10/9	FFHET	-	0.004
Sexual abuse	20/0	7/4	15/4	FFHET	-	0.010
Mental neglect	20/0	9/2	5/14	FFHET	-	<0.001
Physical neglect	17/3	10/1	17/2	FFHET	-	1.00
Viol + SUD + CRIM	17/3	7/4	11/8	FFHET	-	0.172
Total number of all episodes	0.0	1.45 (0.52) ^c^	2.47 (0.90) ^b^	KW	-	<0.001
Lifetime suicidal behaviors	−0.987 (0.0) ^b,c^	0.044 (0.613) ^a,c^	1.013 (0.611) ^a,b^	KW	-	<0.001
Reoccurrence of illness	−1.084 (0.00) ^b,c^	0.170 (0.353) ^a,c^	1.042 (0.429) ^a,b^	KW	-	<0.001
Current suicidal behaviors	−0.916 (0.0) ^b,c^	0.082 (0.789) ^a,c^	0.917 (0.762) ^a,b^	KW	-	<0.001
PC_phenome	−1.149 (0.329) ^b,c^	0.435(0.262) ^a,c^	0.940 (0.354) ^a,b^	209.16	2/47	<0.001

Results are shown as mean ± SD. F: results of analysis of variance; χ^2^: analysis of contingency tables, FFHET: Fisher–Freeman–Halton Exact Test, KW: Kruskal–Wallis test. ^a,b,c^: pairwise comparisons among group means; BMI: body mass index; HAMD: Hamilton Depression Rating Scale score; STAI: Spielberger State and Trait Anxiety, State version; ACEs: adverse childhood experiences; TUD: tobacco use disorder; Viol + SUD + CRIM: violence, substance use disorders or criminal acts in family; PC_phenome: first principal component extracted from severity of depression, anxiety and suicidal behavior data.

**Table 3 brainsci-13-01090-t003:** Measurements of the growth factors and the ratios in major depressed (MDD) patients versus healthy controls (HC).

Variables (z Scores)	HC	MDD	F	df	*p*
NGF	0.539 ± 0.224	−0.359 ± 0.178	10.15	1/46	0.003
SCF	0.309 ± 0.241	−0.206 ± 0.192	0.82	1/46	0.369
SCGF	0.334 ± 0.226	−0.223 ± 0.180	0.69	1/46	0.412
HGF	0.335 ± 0.224	−0.223 ± 0.178	0.43	1/46	0.515
M-CSF	0.211 ± 0.224	−0.141 ± 0.178	0.14	1/46	0.907
z_growth factor (GF)	−0.450 ± 0.712	0.300 ± 1.060	7.82	1/46	0.008
z_neurotoxicity (NT)	−0.374 ± 0.215	0.249 ± 0.175	5.00	1/46	0.030
zGF_zNT	0.571 ± 0.698	−0.381 ± 0.997	16.63	1/46	<0.001
zNGF_zNT	0.595 ± 0.197	−0.397 ± 0.161	17.23	1/46	<0.001

Shown are the estimated marginal mean values (SE) obtained via univariate general linear model analyses with age and sex as covariates. NGF: nerve growth factor; SCF: stem cell factor; SCGF: stem cell growth factor; HGF: hepatocyte growth factor; M-CSF: macrophage colony stimulating factor. z growth factors: composite built using platelet-derived growth factor, vascular endothelial growth factor, and fibroblast growth factor; neurotoxicity: see Appendix A for computation.

**Table 4 brainsci-13-01090-t004:** Intercorrelation matrix between the growth factor variables and adverse childhood experiences (ACEs), recurrence of illness (ROI), suicidal behaviors (SB) and depression phenome (PC_phenome) data.

Variables	ACEs	ROI	Current_SB	PC_Phenome
NGF	−0.417 (0.003)	−0.454 (0.001)	−0.418 (0.003)	−0.424 (0.003)
SCF	−0.198 (0.177)	−0.176 (0.231)	−0.073 (0.620)	−0.156 (0.290)
SCGF	−0.156 (0.288)	−0.105 (0.476)	−0.015 (0.919)	−0.111 (0.453)
HGF	−0.178 (0.227)	−0.129 (0.381)	−0.012 (0.936)	−0.087 (0.555)
M-CSF	−0.111 (0.454)	−0.117 (0.429)	0.01 (0.965)	0.059 (0.692)
z Growth factors (GF)	0.446 (0.001)	0.363 (0.011)	0.249 (0.087)	0.375 (0.009)
zNGF-zGF	−0.552 (<0.001)	−0.522 (<0.001)	−0.426 (0.003)	−0.511 (<0.001)
Neurotoxicity (NT)	0.341 (0.018)	0.274 (0.060)	0.131 (0.376)	0302 (0.037)
zNGF-zNT	−0.536 (<0.001)	−0.514 (<0.001)	−0.387 (0.007)	−0.514 (<0.001)
zNGF-z(GF + NT)	−0.577 (<0.001)	−0.528 (<0.001)	−0.413 (0.004)	−0.522 (<0.001)

Shown are the partial regression analysis coefficients (with exact *p* value), with age and sex as covariates. NGF: nerve growth factor; SCF: stem cell factor; SCGF: stem cell growth factor; HGF: hepatocyte growth factor; M-CSF: macrophage colony stimulating factor. z growth factors: composite built using platelet-derived growth factor, vascular endothelial growth factor, and fibroblast growth factor; z neurotoxicity: see Appendix A for computation.

**Table 5 brainsci-13-01090-t005:** Results of multiple regression analyses with clinical scores of current and lifetime (LT) suicidal behaviors and the depression phenome score (PC_phenome) as dependent variables.

Dependent Variables	Explanatory Variable	β	t	*p*	F Model	df	*p*	R^2^
**#1a. Current_SB**	**Model**				9.08	1/48	0.004	0.159
	zNGF-zGF	−0.399	−3.01	0.004				
**#2a. LT_SB**	**Model**				13.00	1/48	<0.001	0.197
	zNGF-zNT	−0.463	−3.61	<0.001				
**#2b. LT + Current_SB**	**Model**				11.20	1/47	0.002	0.189
	zNGF-zNT	−0.435	−3.35	0.002				
**#3a. PC_phenome**	**Model**				14.64	2/47	<0.001	0.384
	zNGF-z(NT + GF)	−0.563	−477	<0.001				
	Age	−0.280	−0.39	0.021				
**#3b. PC_phenome**	**Model**				27.32	4/45	<0.001	0.708
	Mental trauma	0.580	6.06	<0.001				
	Sexual trauma	0.414	4.96	<0.001				
	zNGF-zNT	−0.271	−2.88	0.006				
	Age	−0.197	−2.40	0.020				

NGF: nerve growth factor; zGF: composite built using platelet-derived growth factor, vascular endothelial growth factor, and fibroblast growth factor; zNT: neurotoxicity composite: see Appendix A for computation.

## Data Availability

The dataset generated during and/or analyzed during the current study will be available from the corresponding author (M.M.) upon reasonable request and once the dataset has been fully exploited by the authors.

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
