# Peer review of "Lower Nerve Growth Factor Levels in Major Depression and Suicidal Behaviors: Effects of Adverse Childhood Experiences and Recurrence of Illness"

_brainsci, 2023, doi:10.3390/brainsci13071090_

Round 1

Reviewer 1 Report

The study related to major depressive disorder (MDD) and its severe subtype, major dysmood disorder (MDMD), are known linked to inflammation and growth factor imbalances. It was claimed that nerve growth factor (NGF) plays a crucial role in neuro-immune communication and may regulate inflammation. Also, the study found that adverse childhood experiences (ACE) and illness recurrence (ROI) are associated with lower NGF levels and decreased NGF and increased neurotoxic cytokines contribute to the severity of depression and chronic inflammation. This study is very interesting in terms of depression and physiological player relationship investigations. On the other hand, there are several questions that should be answered.  

1- How the authors decided that 30 patient sample was enough for the study. Please explain.

2- What was the common drug application for patient groups and was there any biological relationship between drug applications and growth factor expressions?

3- Is it possible for the patient have another pathological condition like diabetes because it was mentioned that patients have variable BMI ratios?

4- Whole RNAseq would be more suitable for these types of studies. Why did the author choose limited numbers of gene panel for the study?

Author Response

How the authors decided that 30 patient sample was enough for the study. Please explain.

@@ANSWER: It is not 30 patients but 50 participants. We performed a priori power analysis to estimate the number of participants based on the knowledge that a large part (10-20%) of the variance in NGF is determined by depression. So, if the alternative hypothesis, based on the known effect size, would not be true, we would not have found significant effects using a sample of n=50 and power 0.8. Furthermore, post-hoc power analysis showed that we worked at a much higher power. It is true that using a bigger study group would improve the parameter estimates, but that is the aim of the study. It is misunderstood that one need huge samples to detect “differences”. Larger studies may disclose smaller effect sizes, say 1% explained variance. But from a clinical point of view, I am not interested at all in the 1% explained variance. If not more than 10% it is not clinically relevant. Furthermore, the primary aim of the study was to examine associations between depression features and NGF.     

What was the common drug application for patient groups and was there any biological relationship between drug applications and growth factor expressions?

@@ANSWER: it was and is explained that: We statistically accounted for the potential effects of the patients' drug use, including benzodiazepines (n=22), atypical antipsychotics (n=14), sertraline (n=18), other antidepressants (n=8, including fluoxetine, venlafaxine, escitalopram, bupropion, and mirtazapine), and mood stabilizers (n=4). Nevertheless, no significant drug effects on the GF could be detected.

3- Is it possible for the patient have another pathological condition like diabetes because it was mentioned that patients have variable BMI ratios?

@@ANSWER: It is explained in the methods section that we excluded subjects with diabetes type 1 and type 2. Furthermore, as explained in the results, there were no effects of BMI on the primary outcome variables.

4- Whole RNAseq would be more suitable for these types of studies. Why did the author choose limited numbers of gene panel for the study?

@@ANSWER: I do not understand: the aim of the study was to measure NGF protein levels, not mRNA or DNA. It would be interesting to do this, of course, but personally I am always more interested in protein levels because the technical and analytical variability are low enough (especially when we use only one plate) to be used in precision medicine analyses (as we did also in the current study). I am not inclined to use RNAseq data in such an approach.

Reviewer 2 Report

The manuscript submitted for review attempts to link the NGF to adverse childhood experiences. For this reason, patients' blood was subjected to the procedure and cytokines and growth factors were assessed on a LUMINEX apparatus. Various statistical analyses were then performed to relate the different factors.

The reviewed manuscript, despite its interesting premise, was written in a rather complicated way.

Already in the introduction there are analyses of the results which, instead of explaining the reason for the research undertaken, unfortunately obscure it. It should be considered whether it would not be better to move the analyses to the results, which could structure the manuscript and make it easier to understand.

In addition, comprehension is not facilitated by the very large number of abbreviations used, which often forces one to interrupt the manuscript being read and seek clarification. If it is acceptable in a journal, a list of abbreviations should be considered.

In section 3.3 of the results, it is not easy to see what the authors wanted to convey through the analysis. It would have been helpful to define the purpose already in the headline and to have a clearer message in the text.

The discussion is well-written and just what the authors were aiming for is clear from it.

Author Response

The reviewed manuscript, despite its interesting premise, was written in a rather complicated way.

@@ANSWER: I do not think there is anything complicated. It is all very straightforward for people who know multiple regression and factor analysis. If one does not know these techniques it is impossible to understand and it is best not to try to read the Results section. That is why I repeat the main findings in the discussion.

Already in the introduction there are analyses of the results which, instead of explaining the reason for the research undertaken, unfortunately obscure it. It should be considered whether it would not be better to move the analyses to the results, which could structure the manuscript and make it easier to understand.

@@ANSWER: These are preliminary analyses that were performed to delineate the specific aims of the study. As such they cannot be placed in the Results (preliminary analyses belong to the background). In addition, I do not understand why this would be difficult to comprehend: I just give a list of functions (based on PPI analysis) of the GFs included. So, table 1 is quite easy to comprehend even to people who do not know PPI annotation analysis.

In addition, comprehension is not facilitated by the very large number of abbreviations used, which often forces one to interrupt the manuscript being read and seek clarification. If it is acceptable in a journal, a list of abbreviations should be considered.

@@ANSWER: we added a list of abbreviations (see placed after the Abstract).

In section 3.3 of the results, it is not easy to see what the authors wanted to convey through the analysis. It would have been helpful to define the purpose already in the headline and to have a clearer message in the text.

@@ANSWER: now we made two subheadings 3.3. and 3.4, and 3.4 conveys what is done: 3.4. Results of multiple regression analysis with clinical data as dependent variables. In fact, it is all very simple if one knows multiple regression analyses.

The discussion is well-written and just what the authors were aiming for is clear from it.

@@ANSWER: thank you.

Round 2

Reviewer 1 Report

There is nothing to add.